# Cell-Free DNA Analysis by Whole-Exome Sequencing for Hepatocellular Carcinoma: A Pilot Study in Thailand

**DOI:** 10.3390/cancers13092229

**Published:** 2021-05-06

**Authors:** Pattapon Kunadirek, Natthaya Chuaypen, Piroon Jenjaroenpun, Thidathip Wongsurawat, Nutcha Pinjaroen, Pongserath Sirichindakul, Intawat Nookaew, Pisit Tangkijvanich

**Affiliations:** 1Center of Excellence in Hepatitis and Liver Cancer, Department of Biochemistry, Faculty of Medicine, Chulalongkorn University, Bangkok 10330, Thailand; pkunadirek@gmail.com (P.K.); natthaya.ch56@gmail.com (N.C.); 2Department of Biomedical Informatics, College of Medicine, University of Arkansas for Medical Sciences, Little Rock, AR 72205, USA; piroon.jen@mahidol.edu (P.J.); thidathip.won@mahidol.edu (T.W.); 3Division of Bioinformatics and Data Management for Research, Department of Research and Development, Faculty of Medicine Siriraj Hospital, Mahidol University, Bangkok 10700, Thailand; 4Department of Radiology, Faculty of Medicine, Chulalongkorn University, Bangkok 10330, Thailand; fon_nutcha@hotmail.com; 5Department of Surgery, Faculty of Medicine, Chulalongkorn University, Bangkok 10330, Thailand; boonchoog1@gmail.com

**Keywords:** hepatocellular carcinoma, cell-free DNA, whole-exome sequencing, biomarker, Thailand, Oxford Nanopore Technologies, copy-number variants

## Abstract

**Simple Summary:**

Liquid biopsy for cell-free DNA (cfDNA) is a non-invasive technique to characterize the genetic profile of a tumor. Despite being a valuable tool, there is no mutational profile of cfDNA from hepatocellular carcinoma (HCC) in patients from Thailand, where HCC is prevalent. The present study aimed to demonstrate the utility of using whole-exome sequencing of cfDNA to define the somatic mutation profiles of HCC in Thai patients who underwent nonoperative therapies. The level of cfDNA was higher in HCC patients than in chronic hepatitis patients. Single nucleotide variations were present in somatic genes in cfDNA, including in *ZNF814*, *HRNR*, *ZNF492*, *ADAMTS12*, *FLG*, *OBSCN*, *TP53*, and *TTN*. The co-occurrence of *HRNR* and *TTN* mutations in cfDNA was associated with shorter overall survival. These findings indicate that the mutational profiles of cfDNA reflected those of HCC tissue, and cfDNA could serve as a useful biomarker for diagnosis and prognosis in HCC patients.

**Abstract:**

Cell-free DNA (cfDNA) has been used as a non-invasive biomarker for detecting cancer-specific mutations. However, the mutational profile of cfDNA in Thai patients with hepatocellular carcinoma (HCC) has not been investigated. Here, we demonstrated the utility of using whole-exome sequencing (WES) of cfDNA to define the somatic mutation profiles of HCC in Thai patients. The comprehensive profile of cfDNA was determined with WES to identify variants in matched cfDNA and germline DNA from 30 HCC patients in Thailand who underwent nonoperative therapies. The level of cfDNA was higher in HCC patients compared with chronic hepatitis patients (*p*-value < 0.001). Single nucleotide variants were present in somatic genes in cfDNA, including in *ZNF814* (27%), *HRNR* (20%), *ZNF492* (20%), *ADAMTS12* (17%), *FLG* (17%), *OBSCN* (17%), *TP53* (17%), and *TTN* (17%). These same mutations were matched to HCC mutation data from The Cancer Genome Atlas (TCGA) and a previous Thai HCC study. The co-occurrence of *HRNR* and *TTN* mutations in cfDNA was associated with shorter overall survival in HCC patients (hazard ratio = 1.60, *p*-value = 0.0196). These findings indicate that the mutational profile of cfDNA accurately reflected that of HCC tissue and suggest that cfDNA could serve as a useful biomarker for diagnosis and prognosis in Thai HCC patients. In addition, we demonstrated the use of the pocket-sized sequencer of Oxford Nanopore Technology to detect copy-number variants in HCC tissues that could be applied for onsite clinical detection/monitoring of HCC.

## 1. Introduction

Hepatocellular carcinoma (HCC) is the most common type of liver cancer (80% of liver cancer). Liver cancer is considered the sixth most common cancer and the second leading cause of death worldwide. A high incidence of HCC has been observed in Eastern Africa and Southeast Asia, including Thailand [1]. HCC frequently develops in the context of underlying chronic liver diseases such as chronic hepatitis or cirrhosis. Most HCC patients are diagnosed at an advanced stage of HCC and experience a short survival time after diagnosis. However, early HCC diagnosis can improve survival due to the efficacy of therapeutic approaches—resection and transplantation are effective therapeutics, but only for early-stage HCC [2]. Therefore, early diagnosis is the key to a good prognosis. Currently, serum alpha-fetoprotein (AFP) is the most widely used biomarker for HCC screening. However, serum AFP assay has a low sensitivity (62.4%) with a high false-negative rate for early HCC diagnosis [3].

Cell-free DNA (cfDNA) consists of short fragments of double-stranded DNA, with lengths ranging from 160 to 200 base pairs. It can be released into plasma from apoptotic or necrotic tumor cells as circulating tumor DNA [4]. Therefore, cfDNA has been used for the non-invasive diagnosis of cancers to provide comprehensive information regarding cancer-associated genetic profiles such as single nucleotide variants (SNVs), copy number variants (CNVs), and epigenetic patterns [4]. In previous studies, the potential utility of cfDNA levels and mutations as a potential clinical biomarker for HCC were investigated, as reviewed by Howell et al. [5]. Based on the eight genes associated with HCC identified from the COSMIC database [6], Howell et al. [7] reported mutations in *ARID1A* (11.7%), *CTNNB1* (7.8%), and *TP5*3 (7.8%) to occur frequently in HCC cfDNA in a European population. The somatic mutations in cfDNA, including SNVs and CNVs, were used to monitor for HCC recurrence in a long-term followup study of a Chinese population [8]. cfDNA was considered a secondary alternative to tumor biopsy to observe genomic alterations of intratumoral heterogeneity (ITH) in HCC that may preclude the need for repeated biopsies [9,10]. Recently, it was shown that the mutated genes of cfDNA represented cancer-associated genes in 63% (19/30) of patients, and cfDNA could be used for tumor genetic profiling when a biopsy is unavailable [11]. These studies indicated that cfDNA could be a tumor marker for diagnostic and real-time malignancy monitoring to help adjust or guide treatment plans. However, the utility of cfDNA quantification and somatic mutation detection for HCC in a Thai population has not been investigated at the genome-wide level.

HCC is highly heterogeneous in terms of genome composition and genes mutated [12]. This malignancy commonly presents with the molecular anomalies of mutations in the *TERT* promoter (60% of the patients in the study), *TP53* (35–50%), *CTNNB1* (20–40%), *AXIN1* (9–13%), *LAMA2* (5–12%), *ARID1A* (12%), *WWP1* (9%), and *RPS6KA3* (8%) genes [13]. However, ethnicity could contribute to global differences in the molecular profile of HCC due to the presence of various risk factors such as hepatitis B virus, hepatitis C virus, alcohol, and metabolic syndrome [14]. Some somatic mutations of HCC in Thailand were consistent with those identified in COSMIC HCC data, but many mutated genes in Thai HCC patients were not found in the COSMIC data [15]. Thus, it is important to characterize the mutational profiles of cfDNA from HCC patients in Thailand even though previous studies of HCC cfDNA have been performed [7,8,9,11,16,17]. Further, most studies performed targeted sequencing (~140 genes), thus missing many potential mutations that might be important in HCC. Therefore, we reasoned that whole-exome sequencing (WES) could provide more comprehensive data to investigate the mutational landscape of cfDNA.

In this study, we used WES to investigate the somatic mutational profile of cfDNA from matched cfDNA and peripheral blood mononuclear cells (PBMCs) from HCC patients in Thailand and demonstrated the utility of cfDNA for potential clinical application as a non-invasive diagnostic and prognostic marker for HCC. We also performed a pilot study using Oxford Nanopore Technology sequencing to detect CNVs in HCC tissues that might be applied for onsite clinical detection/monitoring of HCC [18].

## 2. Results

### 2.1. Patient Characteristics and cfDNA Quantification

This study included 60 patients with HCC who underwent nonoperative therapies and 17 patients with chronic hepatitis (CH). The patients’ characteristics are summarized in Appendix A. The mean age of patients in the HCC group was significantly higher than in the CH group (mean age 62.7 ± 10.3 and 54.8 ± 7.6 years, *p*-value = 0.005). Poorer biochemistry parameters were found in the HCC group than in the CH group, including platelet count, direct bilirubin, total bilirubin, serum albumin, aspartate aminotransferase, and alanine aminotransferase (*p*-value ≤ 0.05). To establish a relationship between total plasma cfDNA and patient characteristics, plasma cfDNA levels were quantified before treatment procedures. The levels of cfDNA and serum AFP were significantly higher in the HCC group than in the CH group (mean cfDNA levels 27.4 ± 37.1 and 6.0 ± 3.4 ng/mL, *p*-value < 0.001, Figure 1A). The highest levels of cfDNA were found in patients with advanced-stage HCC (BCLC stage C) compared with early-stage (stage A) and intermediate stage (stage B) (*p*-value = 0.001), and the levels of cfDNA were elevated in HCC patients with ≥5 cm tumor size compared <5 cm tumor size (*p*-value = 0.013) (Figure 1B). In addition, there was a significant positive correlation between platelet count, direct bilirubin, and tumor size (Appendix A). Receiver operating characteristic analysis showed that the area under the curve (AUC) of plasma cfDNA and serum AFP levels was 0.89 and 0.86, respectively, and the combined plasma cfDNA and serum AFP levels increased the ability to distinguish HCC patients from CH patients (AUC = 0.96) (Figure 1C). Further, plasma cfDNA and matched germline DNA from 30 HCC patients were selected for WES according to cfDNA quality; the median amount of cfDNA in these samples was 117.9 ng (ranging from 57.3–1200 ng). The characteristics of these patients are presented in Table 1.

### 2.2. Somatic Mutation Profiling of cfDNA Using Whole-Exome Sequencing

To investigate the genomic profile of cfDNA from the HCC patients in Thailand, the 30 cfDNA and matched germline DNA samples were subjected to whole-exome sequencing (WES) with a target region of approximately 35.7 Mb. WES was carried out at a median sequencing depth of 55.59× for cfDNA and 57.49× for germline DNA (Appendix A). To identify somatic mutations, germline DNA from PBMCs was used as a control. All samples contained somatic mutations, with a median of 49.5 mutations per sample (ranging from 3–818 mutations) (Appendix A). The top 25 most frequently mutated genes were found to be mutated in more than 10% of patients for each gene and covered 76.67% (23/30) of patients (Figure 1D). Interestingly, the greatest number of mutations were found in early-stage HCC compared with other stages. We also found that missense variants were the most frequent mutation (Appendix A). A base transition of nucleotide changes (C > T and T > C) dominated the mutation spectrum (Appendix A), which is comparable to previous reports [19]. C > T transition was associated with mismatch repair deficiency, and T > C was associated with alcohol consumption in HCC.

In our cohort, we found many mutations that potentially disrupted oncogenic pathways, including RTK-RAS (36.67%, 11/30 patients), WNT (33.33%, 10/30 patients), NOTCH (36.67%, 11/30 patients), and Hippo pathways (40%, 12/30 patients) (Figure 2A). The eight highly mutated genes in more than 15% of patients were ZNF814 (27%, 8/30 patients), HRNR (20%, 6/30 of patients), ZNF492 (20%, 6/30 patients), ADAMTS12 (17%, 5/30 patients), FLG (17%, 5/30 of patients), OBSCN (17%, 5/30 patients), TP53 (17%, 5/30 of patients), and TTN (17%, 5/30 patients). In addition, we found that ZNF814 and ZNF492 were mutated at a single location; whereas, other genes contained mutations at multiple locations (Figure 2B). The somatic mutations of HRNR and TTN were found exclusively in patients with early-stage HCC (A) and a low level of serum AFP (Appendix A).

### 2.3. Comparison of Top Frequently Mutated Genes across Studies, and Clinical Application

To investigate and verify the concordance between the top eight frequently mutated genes in plasma cfDNA and HCC tissue in other studies, HCC patient data (TCGA, Firehose Legacy, http://gdac.broadinstitute.org, accessed date on 16 March 2020) and the cBioPortal [20] online tool were used for exploring these mutated genes in HCC tissues. The highly mutated genes identified in our study were also altered in 228 tissue samples from 362 patients with HCC (62.3%). Specifically, genetic alteration of these genes was analyzed and visualized as an oncoprint representing in-frame mutations, missense mutations, truncating mutations, amplifications, and deep deletions along with the race of patients (Figure 3A). Interestingly, we found that OBSCN and FLG were highly mutated in Asian patients compared with White patients (*p*-value = 0.029) (Appendix A).

To examine the concordance between mutations in cfDNA and HCC tissues, the mutation frequencies of these genes in each dataset were examined (Figure 3B). The mutation frequencies of OBSCN and FLG in our study were in considerable agreement with mutation frequencies in HCC tissue from the TCGA dataset. In contrast, the mutation frequencies of ZNF814, ZNF492, and ADAMTS12 were much lower in HCC tissues than in cfDNA. However, different gene mutations can be present based on the race or ethnicity of patients [21]. Therefore, the mutations of ZNF814, ZNF492, and ADAMTS12 might be important for HCC patients from Thailand specifically. To investigate the concordance between mutated gene profiles of cfDNA and tumor DNA from patients with HCC in Thailand, the mutated genes identified in our study were compared with a targeted gene subset analyzed with exome sequencing in another Thai population (564 genes based on commonly mutated genes across various solid tumor types in the COSMIC database [6]) [15]. We found that 49/109 (31%) mutated genes in HCC tissue identified in a previous study overlapped with our data (Figure 3C). Nevertheless, the 560-target exome sequencing dataset in a previous study [15] did not include all eight frequently mutated genes. Thus, it is possible that the targeted exome sequencing missed some mutations compared to WES. In previous studies, the genetic alteration of cfDNA in patients with HCC was investigated using WES [9,22], and the comparison of mutated genes of cfDNA in our study and previous studies found partial concordance (Appendix A). However, these previous studies used a smaller sample size than our study. Moreover, we found co-occurring HRNR and TTN mutations in cfDNA from the same patients; whereas, other mutations were exclusive to each patient (Figure 3D). The prognostic utility of co-occurring mutations was then examined using log-rank analysis and demonstrated by Kaplan–Meier curves with the cBioPortal database. HCC patients with mutations in HRNR and/or TTN experienced shorter overall survival (median = 33.02 months, hazard ratio = 1.60, *p*-value = 0.0196) than HCC patients without those mutations (median = 70.01 months) (Figure 3E).

### 2.4. Copy Number Variants (CNVs) and Oxford Nanopore Application (Pilot Study)

CNVs can contribute to chromosomal alterations, including amplification or deletion of regions of the genome that influence carcinogenesis [23]. A previous study reported CNVs in HCC tissue that resulted in gains in chromosomes 1q, 5p, 6p, 7q, 8q, 17q, and 20q, and losses in chromosomes 1p, 4q, 6q, 8p, 9p, 13q, 14q, 16p-q, 17p, 21p-q, and 22q [24]. Here, we identified CNVs in cfDNA of HCC patients with WES and found that CNVs resulted in gains in chromosome 1q, 3q, 7q, 8q, 12p, 15q, and 17q and losses in chromosome 5p-q in 5/30 patients compared with germline DNA (Figure 4A and Appendix A). However, the noise signals for copy number counts in cfDNA from WES were high, and there was no standard to detect CNVs in cfDNA from WES. As the HCC patients were under nonoperative therapy (no tumor tissue was available), we compared CNVs derived from tumor dissected from another group of Thai HCC patients (see Appendix A). We performed Oxford Nanopore Technology sequencing to detect CNVs in tumor DNA from five HCC tissues using an amplification-free method (SMURF-seq) and compared the results with our cfDNA analysis. CNV analysis showed gains of chromosome 1p and 8p in sample BLM6 and partially in BLM1 (2/5 patients) (Figure 4B and Appendix A). The CNV profile at chromosome 1q and 8p was similarly found in our cfDNA samples.

## 3. Discussion

In this study, we successfully used WES to analyze the mutational landscape in cfDNA samples from 30 patients with HCC in Thailand. We characterized comprehensive genomic profiles, including cfDNA concentration and genetic alteration (SNVs and CNVs). We also demonstrated that the cfDNA concentration could be used as an alternative biomarker to enhance the efficiency of HCC screening. The most common SNVs in cfDNA were also found in tumor DNA from patients with HCC. Interestingly, the co-occurrence of frequently mutated genes in cfDNA was associated with worse overall survival time in patients with HCC. This study suggests that cfDNA liquid biopsy could be both a useful tool for detecting HCC and also a prognostic marker.

It is hypothesized that cfDNA is released from apoptotic and necrotic cells into the blood circulation [25]. In normal conditions, the clearance of cfDNA is conducted by immune cells. However, in tumor conditions, the clearance of cfDNA is not efficient, leading to an accumulation of cellular debris such as DNA [26]. cfDNA can be detected in both cancer patients and healthy patients, but the levels of cfDNA differ between the two [27]. An increase in cfDNA in the blood circulation was observed primarily in patients with tumoral mass compared with non-tumor patients [27]. In concordance with previous studies in HCC [28,29], the levels of cfDNA were significantly higher in patients with HCC than in patients with CH and were associated with worse clinical parameters, including tumor size and BCLC stage. Specifically, the high levels of cfDNA were found in patients with advanced-stage HCC (BCLC stage C) compared with early-stage (stage A) and intermediate stage (stage B). These results suggest that the level of cfDNA reflects tumor progression to a certain extent. Currently, the conventional biomarker for detecting HCC and its recurrence (serum AFP) has limited sensitivity to detect early HCC and can also be elevated in other disease states. A previous study demonstrated that cfDNA could improve the diagnosis of HCC when combined with serum AFP [30]. In agreement with this report, the combination of plasma cfDNA and serum AFP assays increased the performance of HCC screening over either marker alone. These results imply that cfDNA could increase the efficiency of discriminating HCC patients from non-cancer patients.

In addition to analyzing the cfDNA concentration, we performed WES on cfDNA from patients with HCC and analyzed for genetic alterations that could reflect the genetic profile of the tumor mass [11]. To the best of our knowledge, this is the first study that demonstrated genetic alterations in cfDNA from patients with HCC in Southeast Asia (Thailand), where there is a high incidence of HCC [1]. There have been a few studies that reported genetic alterations in cfDNA from a small number of HCC patients using WES [9,22]. In agreement with these reports, we found the same genes to be frequently mutated, including *TP53* (detected in most of the cancers), *FLG*, *TTN*, and *ADAMTS12*, and WES analysis of cfDNA could be used to detect mutated genes in all HCC patients. When comparing with targeted sequencing of cfDNA, WES analysis provides more comprehensive data on the entire set of mutated genes in samples and does not require previous knowledge of the mutational profile [31]. Importantly, the sensitivity of low variant detection is inverse to the proportion of the size of gene panel to sequencing cost [32]. Although we performed WES to analyze cfDNA, the lowest mutation allele frequency was around 0.6–1% in this study. However, the gene alterations identified in cfDNA from patients with HCC were partially concordant with those identified in another WES study on cfDNA and tumoral tissues from Chinese HCC patients [9]. Interestingly, a previous study of cfDNA without prior knowledge of the mutation profile in biopsy tissues demonstrated that 27% of mutations in cfDNA were present in the biopsy [33]. This was similar to our study in that we found 31% concordance between mutated genes in cfDNA and HCC tissue in Thai patients [15]. Furthermore, although our cohort consisted of cfDNA and germline DNA from patients who underwent nonoperative treatment (meaning we were unable to access tumor tissue), we still found the mutations in cfDNA in concordance with other studies [9,15,22] of HCC tissues. These data indicate that cfDNA data could be used to reflect the tumor genome when tumor DNA is not available [33].

In this study, we found that the eight most frequently mutated genes in cfDNA from HCC were also frequently mutated in HCC tissue from TCGA data, including *TP53* (33%), *TTN* (30%), *FLG* (17%), *OBSCN* (16%), *HRNR* (13%), and *ADAMTS12* (4%). Indeed, previous studies demonstrated that *TP53*, *TTN*, *FLG*, and *OBSCN* were frequently mutated genes in HCC patients [34,35]. Regarding *FLG* and *OBSCN*, these mutations were found in Asian patients with HCC, and *FLG* was altered in Asian patients more than in any other ethnicity [35]. Interestingly, in our study, *ZNF814* and *ZNF492* were also frequently mutated at the same sites in cfDNA. Even though these mutations occurred at low frequency in HCC tissue from TCGA data, a recent finding showed that mutations in the ZNF family are associated with human disease, including cancer [36,37]. Several things could account for the high rates of specific mutations found in the current study. One is that our study was based on an Asian population, but current databases are based primarily on White populations. There are different causes of HCC and different genetic backgrounds, even in Asian countries [12]. On the other hand, the HCC study in Thailand demonstrated that HCC subtypes of different ethnicities were not completely matched between Thai HCC patients and those of other races/ethnicities, and somatic mutations from Thai HCC patients were not entirely in agreement with the COSMIC database [15]. In addition, the co-occurrence of mutations in *HRNR* and *TTN* in our study was associated with a worse prognosis in patients with HCC. Thus, the mutations in cfDNA might be prognostic markers for patients with HCC, but this requires further investigation.

The detection of mutations in plasma cfDNA in HCC provides exciting possibilities for guiding treatment in patients. We identified patients with an activating hotspot mutation to the *CTNNB1* gene in the Wnt/beta-catenin pathway. In previous studies [38,39], it was demonstrated that the mutation of S33C and S37A in *CTNNBB1* might lead to loss of phosphorylation sites in the beta-catenin protein, increasing the expression of *CTNNB1* and dysregulating the Wnt/beta-catenin pathway. In this context, a recent study of 31 patients with HCC treated with an immune checkpoint inhibitor demonstrated that the activation of Wnt/beta-catenin signaling was associated with poor response to therapy and shorter survival [40]. Moreover, a study of 17 regorafenib-treated HCC patients demonstrated that *CTNNB1* mutation was found exclusively in non-responders [41]. Sorafenib is used globally as a standard first-line treatment for advanced HCC and targets multiple kinases, including BRAF, a serine/threonine-protein kinase. In a previous study, we identified patients with *BRAF* mutations, and these correlated with response to the multi-kinase inhibitor sorafenib [42]. Thus, cfDNA profiling may allow the use of precision oncology to improve the efficiency of treatments and ultimately the clinical outcome of patients with HCC.

In CNV analysis, amplifications in chromosomes 1q, 3q, 7q, 8q, 12p, 15q, and 17q and loss in chromosome 5p-q were observed in 16.67% of cfDNA HCC samples (5/30 samples). Even though the analysis of CNVs in cfDNA was complicated by a high background signal in our study (due to the fragmentation of cfDNA and sequencing bias from WES), the CNVs we identified were similar to those identified in previous studies of HCC tissue and cfDNA, such as gains in chromosomes 1q, 7q, 8q, and 17q [43,44]. These CNVs of cfDNA were also used for scoring genomic instability, which was associated with tumor progression and overall survival time in patients with HCC [44]. These data suggest that CNVs of cfDNA could serve as prognostic markers for HCC. Recently, SMURF-seq was developed to improve the efficiency of CNV analysis by concatenating short fragments into long molecules before sequencing [45]. SMURF-seq can be performed with low-coverage reads, shorter time, and at low cost and obtain similar CNV data as short-read sequencing within a day; it also uses a portable device that would be suitable for clinical sites. Therefore, SMURF-seq was used to perform CNV analysis on HCC tissues to compare with CNVs from cfDNA. To our knowledge, this is the first study to perform CNV analysis for HCC using nanopore technology. SMURF-seq clearly identified CNVs in HCC, such as gains in chromosomes 1q and 8q, which are commonly found in HCC cfDNA and tissue [44]. These results indicate that gains in chromosomes 1q and 8q were concordant across HCC cfDNA and tissue DNA, as identified with WES and SMURF-seq. Even though SMURF-seq can give a cursory view of CNVs in HCC tissues, greater sequencing depth is still needed to improve the resolution of CNVs to ensure the reliability of CNV detection.

In conclusion, our study demonstrated the comprehensive analysis of cfDNA from patients with HCC in Thailand using WES and provided a genetic profile of that cfDNA. We conclude that cfDNA could be a biomarker for the diagnosis and prognosis of HCC and may provide SNV and CNV profiles of tumoral tissue, which could guide targeted therapeutic strategies for HCC when tumor tissue is not available. However, our study was limited by the small sample size and the unavailability of paired tumor tissues for WES analysis, which would have allowed direct comparison of the mutational profiles from cfDNA and HCC tissue.

## 4. Materials and Methods

### 4.1. Patients and Study Overview

Patients with HCC or chronic hepatitis (CH) were enrolled from King Chulalongkorn Memorial Hospital between October 2018 and October 2019. HCC patients were diagnosed according to current American Association for the Study of Liver Diseases (AASLD) guidelines by contrast-enhanced imaging technologies (CT or MRI) [46]. Baseline clinical data were collected from all patients, including liver function blood test results, serum alpha-fetoprotein (AFP), and stage of HCC classified by the Barcelona Clinic Liver Cancer (BCLC) staging system [47]. Only HCC patients at stages A to C were included in the cfDNA study. This study was conducted in accordance with the Declaration of Helsinki for the participation of human individuals. Written informed consent was received from all patients, and the protocols were approved by the Institute Ethics Committee of Faculty of Medicine, Chulalongkorn University (IRB No. 313/62).

A total of 193 participants were enrolled in the study, including 171 patients with HCC who underwent nonoperative therapies such as trans-arterial chemoembolization (TACE) and/or radiofrequency ablation (RFA) or microwave ablation (MWA), which are the common treatments for intermediate- and advanced-stage HCC (stages B and C). In addition, 5 patients who underwent hepatic resection and 17 patients with CH were recruited for CNV analysis using Oxford Nanopore Technologies (ONT) sequencing. The exclusion criteria for patients with HCC who underwent nonoperative therapy included age, non-first treatment, recurrence of HCC, liver metastasis of other cancer types, and HIV infection, resulting in 60 patients with HCC.

Several steps were taken to produce a comprehensive profile of cfDNA. First, the concentration of plasma cfDNA was evaluated in the HCC and CH groups. Then, 30 cfDNA samples from HCC patients were selected based on sample quality and purity to perform WES. The genetic alterations in the cfDNA were identified by comparing the mutational profile with that of germline DNA from individual patients; mutations of interest included SNVs and CNVs. SNVs from our cfDNA samples were compared with samples from other studies. Additionally, we performed ONT sequencing to analyze CNVs in HCC tissues from 5 patients who underwent hepatic resection. The study overview is shown in Appendix A.

### 4.2. Sample Collection and DNA Extraction

Twelve milliliters of blood were collected in EDTA tubes from patients with HCC and CH. For HCC, blood samples were obtained before nonoperative therapy (At the time of sample collection, the patients were not treated surgically; therefore, the tumor tissue is not available). Plasma and Peripheral blood mononuclear cells (PBMCs) were isolated from blood samples from each patient (Both HCC and CH patients). Plasma was purified within 3 h by 2 steps of centrifugation at 1600× *g* for 10 min to separate plasma from other blood components and 16,000× *g* for 10 min at 4 °C to discard debris from plasma, respectively, and stored at −80 °C until use to reduce cfDNA degradation. PBMCs were isolated from the same patients by density centrifugation (Ficoll-Paque) and stored at −20 °C. Additionally, 5 HCC tissue samples were obtained from other patients with HCC at hepatectomy; HCC was confirmed by histopathology. Liver tissues were stored immediately at −80 °C.

The cfDNA was extracted from 6 mL plasma using the QIAamp MinElute cfDNA Kit (Qiagen) following the manufacturer’s instructions. Germline DNA and tumor DNA were extracted from PBMCs and liver tissues using the GenUP germline DNA Kit (Biotechrabbit, Berlin, Germany) according to the manufacturer’s instructions. Qubit dsDNA HS Assay Kit (Invitrogen, Waltham, MA, USA) was used to quantify DNA. The quality of DNA specimens and cfDNA fragments were assessed by 2% gel electrophoresis before library preparation.

### 4.3. Library Preparation and Whole-Exome Sequencing (WES)

A total of 30 paired cfDNA and germline DNA samples were sequenced by Novogene Bioinformatics Technology Co. Ltd. (Beijing, China). An input of 25–50 ng cfDNA and germline DNA samples were used for exome capturing and library construction using SureSelectXT *Homo sapiens* All Exon V6 + UTR Kit (91 Mb) (Agilent Technologies, Santa Clara, CA, USA). Briefly, germline DNA was fragmented to 150 bp by acoustic fragmentation (Covaris, Woburn, MA, USA). Unfragmented cfDNA and fragmented germline DNA were subjected to end repair, A-tailing, and adapter ligation, exome hybrid capture, and PCR amplification. Equimolar library pools were sequenced on an Illumina HiSeq X-Ten platform, generating 150 bp paired end reads. The average coverage of WES (cfDNA and germline DNA) was approximately 56×.

### 4.4. WES Data Analysis

Raw FASTQ files were processed to remove adapters and low-quality reads (quality score < Q30) using Trimmomatic version 0.36 with default parameters [48]. Pre-processing steps were performed using the nfcore/sarek pipeline version 2.5.1 [49] to generate BAM files according to Genome Analysis Toolkit (GATK) best practices [50]. In brief, high-quality reads were aligned and mapped to a human reference genome (GRCh37) with BWA-MEM version 0.7.17 [51]. Duplicated mapped reads were marked, and the base quality score was recalibrated to obtain more accurate bases using GATK version 4.1.2.0. After base recalibration, BAM files were used for SNV and Indel calling with Mutect2 [52]. A panel of normals was created from germline DNA samples for filtering out variants in cfDNA. Somatic variants from cross-contamination between samples and sequencing artifacts were also calculated and filtered out. Functional variants annotation was accessed by Funcotator. Variants with low-quality reads < 30, depth coverage < 20, or <2 reads in cfDNA were filtered out. For CNV analysis, BAM files without reading duplicate marking and base recalibration were used for CNV analysis with CNVkit version 0.9.0 [53] and Ginkgo [54] with default settings. Gain or loss of copy number variation was identified using absolute log 2 of ratios > 0.2 as a cut-off. Annotated somatic variations were analyzed and visualized by MAFtools R package version 2.3.30 [55]. The script for WES analysis is provided in Appendix A.

### 4.5. Library Preparation and Sequencing on an ONT Platform

Five tumor DNA samples were sequenced on a MinION sequencing device (Oxford Nanopore Technologies, ONT). CNVs in tumor DNA were assessed using the SMURF-seq protocol as described previously [45]. More specifically, tumor DNA was fragmented into short fragments with Anza 64 SaqAI restriction enzyme (Thermo Fisher, Waltham, MA, USA) then randomly ligated to form a long DNA fragment using Anza T4 DNA Ligase Master Mix (Thermo Fisher, Waltham, MA, USA). Then, a rapid barcoding kit (SQK-RBK004, ONT) was used for library preparation according to the manufacturer’s protocol. Sequencing of the tumor DNA was performed on a single R9.4/FLO-MIN106 flow cell (ONT) on a MinION Mk1B.

Raw data from sequencing was generated with MinKNOW software version 1.7.14 (ONT) and converted to FAST5 files that were used for base calling with filter quality read score > 8; data was then de-multiplex barcoded with Guppy version 2.3.4 software (ONT) into FASTQ files. Then, reads were mapped to a human reference genome (GRCh37) with Minimap2 version 2.17 software [56], and BAM files were created with Samtools version 1.10 [57]. BAM files were sorted and converted to BED format using bamtobed from Bedtools package version 2.25. The BED files were used as an input file for Gingko [54] to perform CNV analysis for each sample.

### 4.6. Statistical Analysis

Statistical analysis was performed using GraphPad Prism version 7 for Windows. The concentration values of cfDNA are presented as the mean ± standard deviation and were used for comparison between groups by Student’s unpaired *t*-tests. Receiver operating characteristic analysis was performed, and *p*-value ≤ 0.05 was considered statistically significant.

## 5. Conclusions

Our study demonstrated that comprehensive profiles of patients with HCC could be generated from cfDNA using WES, potentially serving as a diagnostic and prognostic biomarker. Mutational analysis of cfDNA could also be used to design personalized medicine approaches for patients with HCC.

## Figures and Tables

**Figure 1 cancers-13-02229-f001:**
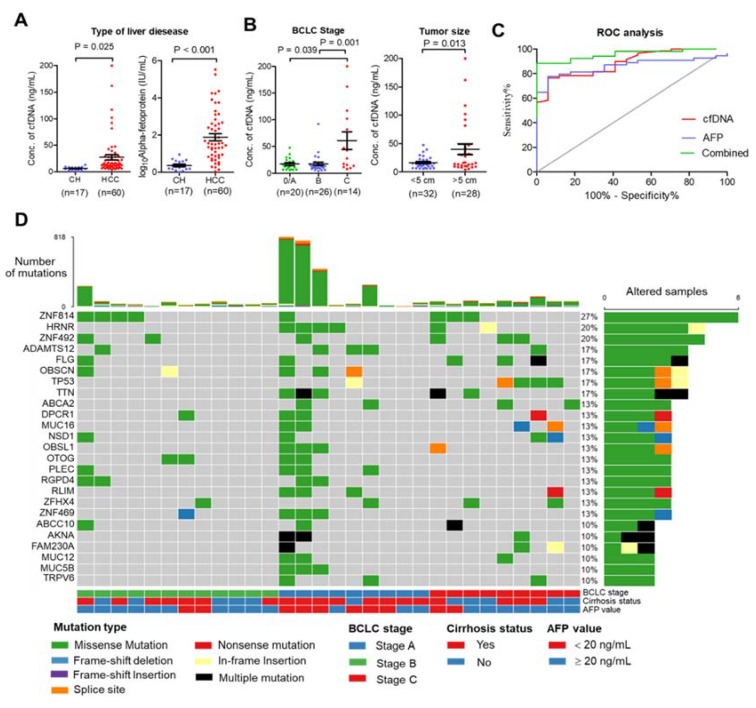
Clinical relationship and landscape of somatic alterations detected in cfDNA from patients with HCC. (**A**) Plasma cfDNA (left) and AFP (right) were significantly higher in HCC patients than in chronic hepatitis patients (CH). (**B**) Relationship of cfDNA and clinical data. Plasma cfDNA was significantly higher in stage C HCC than in stage A or B (left), and plasma cfDNA was higher in HCC patients with tumor size > 5 cm than with tumor size < 5 cm. (**C**) Diagnostic value of cfDNA, serum AFP, and combined cfDNA and AFP. (**D**) Landscape plot of 25 most frequently mutated genes in 30 HCC cfDNA samples. Genes are ordered by mutation frequency, and samples are ordered according to BCLC stage, cirrhosis status, and AFP value as indicated in annotation (bottom). The top bar shows the number of mutations for each sample. The sidebar shows the number of altered samples for each gene.

**Figure 2 cancers-13-02229-f002:**
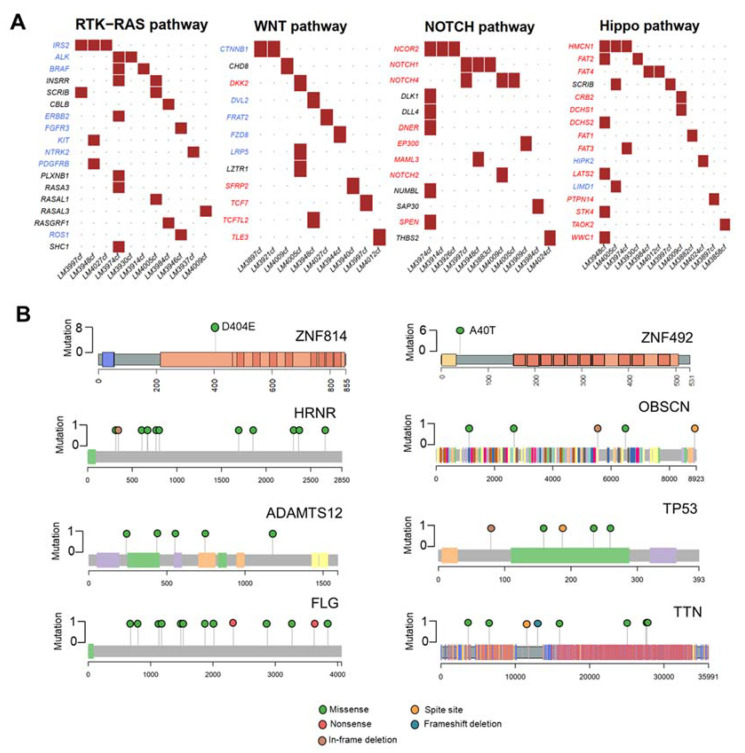
Mutational analysis of the 30 HCC cfDNA samples. (**A**) Mutated genes in HCC cfDNA related to oncogenic pathways. Oncogenes are highlighted in red; tumor suppressor genes are highlighted in blue. (**B**) Lollipop plots displaying mutation distribution and protein domains of the top eight frequently mutated genes and demonstrating the locations of *ZNF814* and *ZNF492* mutations.

**Figure 3 cancers-13-02229-f003:**
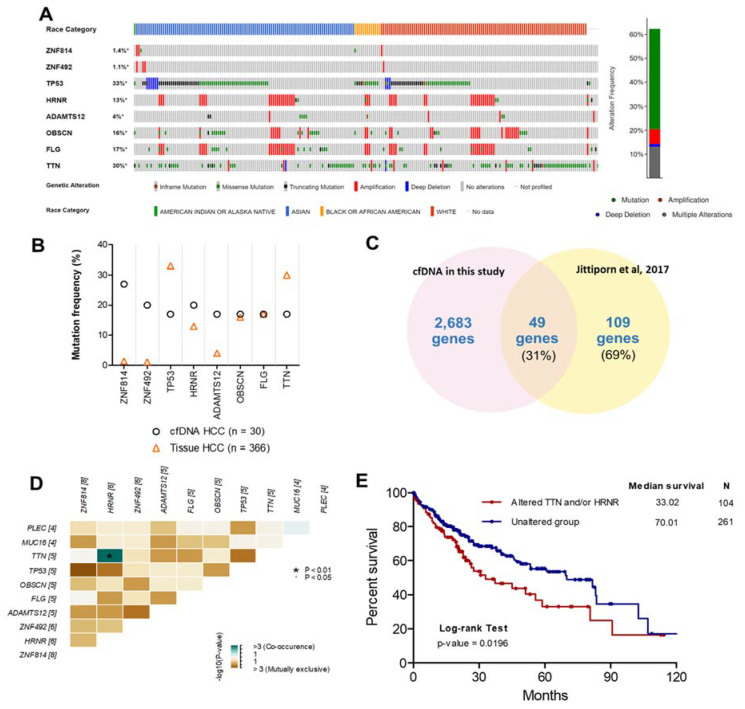
Comparison of the top eight frequently mutated genes from other studies. (**A**) Oncoprint of eight highly mutated genes in 366 HCC patients using the cBioPortal dataset ordered by the race of patients and type of genetic alteration. Far-right: alteration frequency (percentage) of patients with an alteration. (**B**) Comparison of mutation frequency between cfDNA in the present study and tissue DNA from the TCGA dataset. (**C**) Intersect of mutated genes between cfDNA in the present study and HCC tissue DNA from exome sequencing data (Thailand). Percentages are the proportion of mutated genes in HCC tissue DNA (Thailand). (**D**) Co-occurrence of 10 mutated genes. Green indicates a tendency toward co-occurrence. (**E**) Overall survival analysis of patients with HCC using the TCGA dataset for *TTN* and/or *HRNR* mutation.

**Figure 4 cancers-13-02229-f004:**
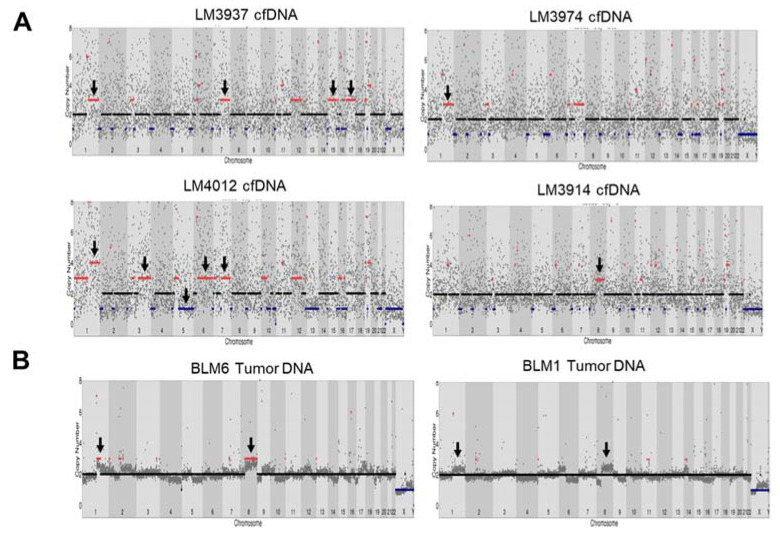
Copy number variants detection in HCC cfDNA and tumor DNA using whole-exome sequencing (WES) and Oxford nanopore sequencing. (**A**) CNVs in HCC cfDNA from LM3937, LM3974, LM4012, and LM3914 using WES. (**B**) CNVs in tumor DNA using Oxford nanopore sequencing.

**Table 1 cancers-13-02229-t001:** Baseline characteristics of cfDNA from 30 patients with HCC for whole-exome sequencing.

Baseline Characteristics	HCC (*n* = 30) *
Sex (male, %)	25 (83.33)
Age, years	64.93 (51–86)
Laboratory data	
Aspartate aminotransferase, IU/L	67.93 (11.0–151.0)
Alanine aminotransferase, IU/L	57.5 (11.0–152.0)
Serum albumin, g/dL	3.5 (2.2–4.4)
Total bilirubin, mg/dL	1.2 (0.3–2.2)
Platelet, 10^9^/L	190.6 (28.0–685.0)
Alpha-fetoprotein, IU/mL	8102.9 (0.9–179,249.0)
Liver disease status	
HBV infection	10 (33.33)
HCV infection	7 (23.33%)
HBV and HCV infection	1 (3.33%)
Non-viral infection	12 (40.00%)
Cirrhosis	19 (63.33%)
BCLC stage	
A	9
B	12
C	9
Tumor number	
Single	19
Multiple	11
Tumor size, cm	
<5	14
≥5	16

* mean (min–max) or count (%).

## Data Availability

All sequencing data generated in this study are available in the NCBI SRA database under Bioproject number PRJNA713009.

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
