# Peer review of "Cell-Free DNA Analysis by Whole-Exome Sequencing for Hepatocellular Carcinoma: A Pilot Study in Thailand"

_cancers, 2021, doi:10.3390/cancers13092229_

Round 1

Reviewer 1 Report

The authors present a study wherein whole exome sequencing was performed to characterize the mutational profile of cell free DNA in patients with hepatocellular carcinoma (HCC) treated at a tertiary center in Thailand. Considering the higher incidence of HCC in the South Asian population, and the lack of tumor mutational data for non-Caucasian patients, this study provides some unique key findings.

Some specific comments:
Major comments

  1. The authors describe that the whole exome sequencing data regarding the copy number variations of the cell free DNA was noisy. This is indeed a common problem encountered with sequencing of cell free DNA and is also encountered while deciphering data on mutational status. In this scenario why did the authors additionally include surgically rejected hepatocellular carcinomas, when the rest of the study deals with hepatocellular carcinoma not treated surgically, it’s not entirely clear.
  2. If whole exome sequencing would have been performed also on hepatocellular carcinoma tissues, it would have a loud comparison of the mutational profiles of tumor tissue with that of cell free DNA. The authors should address this particular limitation of the study.

Minor Comments 

  1. Abstract; Line 37; the authors should provide the Hazard ratio.
  2. The manuscript needs extensive editing for the language.
  3. Was there any correlation of the ideology of hepatocellular carcinoma with the mutational profile of cell free DNA?
  4. Discussion; line 244-245; The authors probably mean “prognostic marker“ and not “predictive marker”.
  5. What criteria were used for quality control of the data obtained from whole exam sequencing of the cell free DNA?

Reviewer 2 Report

In this manuscript the authors performed whole exome sequencing (WES) of cell-free DNA (cfDNA) obtained from the blood of 30 HCC patients from Thailand. The mutations identified are observed in HCC samples in TCGA database. However, the mutation frequency is different. It was identified that concordant mutations in HRNR and TTN genes, identified in cfDNA, were associated with decreased overall survival. The authors concluded that cfDNA might be a diagnostic/prognostic marker for HCC patients. There is difference in mutation profile in HCC patients depending on race/ethnicity. In this respect, the authors’ studies are important because cfDNA analysis has not been performed in Thai HCC patients before. The findings are clear and convincing and unravel novel mutation frequency and distribution. The observation that Thai patients have higher mutation frequency in ZNF814 and ZNF492 genes mandates more in-depth analysis of the function of these genes in HCC in future studies. This is a clinically relevant study for Thai HCC patients.

There are grammatical errors throughout the manuscript which needs to be carefully edited.

Round 2

Reviewer 1 Report

The authors have adequately addressed my comments.